# Differential Antibody Response against Conformational and Linear Epitopes of the L1 Proteins from Human Papillomavirus Types 16/18 Is Observed in Vaccinated Women or with Uterine Cervical Lesions

**DOI:** 10.3390/vaccines9050442

**Published:** 2021-05-02

**Authors:** Adolfo Pedroza-Saavedra, Angelica Nallelhy Rodriguez-Ocampo, Azucena Salazar-Piña, Aislinn Citlali Perez-Morales, Lilia Chihu-Amparan, Minerva Maldonado-Gama, Aurelio Cruz-Valdez, Fernando Esquivel-Guadarrama, Lourdes Gutierrez-Xicotencatl

**Affiliations:** 1Centro de Investigación Sobre Enfermedades Infecciosas, Instituto Nacional de Salud Pública, 62100 Cuernavaca, Mexico; apedroza@insp.mx (A.P.-S.); aislinn.perez@uaem.edu.mx (A.C.P.-M.); lchihu@insp.mx (L.C.-A.); mmaldona@insp.mx (M.M.-G.); 2Unidad Académica Químico Biológicas y Ciencias Farmacéuticas, Universidad Autónoma de Nayarit, 36715 Tepic, Mexico; angelica.rodriguez@uan.edu.mx; 3Facultad de Nutrición, Universidad Autónoma del Estado de Morelos, 62100 Cuernavaca, Mexico; azucena.salazarp@uaem.edu.mx; 4Facultad de Medicina, Universidad Autónoma del Estado de Morelos, 62100 Cuernavaca, Mexico; fernando.esquivel@uaem.mx; 5Centro de Investigación en Salud Poblacional, Instituto Nacional de Salud Pública, 62100 Cuernavaca, Mexico; acruz@insp.mx

**Keywords:** cervical cancer, human papillomavirus, L1 protein, VLPs, linear and conformational epitopes, HPV vaccine

## Abstract

Antibodies against the Human Papillomavirus (HPV) L1 protein are associated with past infections and related to the evolution of the disease, whereas antibodies against L1 Virus-Like Particles (VLPs) are used to follow the neutralizing antibody response in vaccinated women. In this study, serum antibodies against conformational (VLPs) and linear epitopes of HPV16/18 L1 protein were assessed to distinguish HPV-vaccinated women from those naturally infected or those with uterine cervical lesions. The VLPs-16/18 were generated in baculovirus, and L1 proteins were obtained from denatured VLPs. Serum antibodies against VLPs and L1 proteins were evaluated by ELISA. The ELISA-VLPs and ELISA-L1 16/18 assays were validated with a vaccinated women group by ROC analysis and the regression analysis to distinguish the different populations of female patients. The anti-VLPs-16/18 and anti-L1-16/18 antibodies effectively detect vaccinated women (AUC = 1.0/0.79, and 0.94/0.84, respectively). The regression analysis showed that anti-VLPs-16/18 and anti-L1-16/18 antibodies were associated with the vaccinated group (OR = 2.11 × 10^8^/16.50 and 536.0/49.2, respectively). However, only the anti-L1-16 antibodies were associated with the high-grade lesions and cervical cancer (CIN3/CC) group (OR = 12.18). In conclusion, our results suggest that anti-VLPs-16/18 antibodies are effective and type-specific to detect HPV-vaccinated women, but anti-L1-16 antibodies better differentiate the CIN3/CC group. However, a larger population study is needed to validate these results.

## 1. Introduction

Cervical cancer (CC) is the fourth-most common cancer worldwide [1]. In developing countries like Mexico, CC occupies the third place, and it represents an important public health problem. In 2018, about 569,800 women were diagnosed with CC around the world, and 84% of them corresponded to underdeveloped countries [1]. Human Papillomavirus (HPV) is the etiological agent of CC, and high-risk HPV types 16 and 18 are the most frequently present in this pathology worthwhile and, also, in Mexico [2,3]. Late proteins L1 and L2 are structural components of the viral capsid. Besides, the L1 protein is able to self-assemble in different cellular systems in vitro to produce highly immunogenic Virus-Like Particles (VLPs), which are used in HPV vaccines [4,5], as they generate a protective humoral immune response [6,7].

The use of neutralization assays such as the PBNA (Pseudovirion-Based Neutralization Assay) is considered the gold standard to evaluate the protective potential of the antibodies induced by the HPV vaccines [8]. However, the complexity of the assay makes it difficult to implement it during clinical trials or large epidemiological studies. For this reason, Dessy and coworkers [9] compared the capacity of the direct ELISA-VLPs against the PBNA test to detect neutralizing antibodies and demonstrated that the direct ELISA-VLPs is an excellent surrogate marker to detect neutralizing activity. This assay has recently been used to measure the efficacy of the vaccine [9,10,11,12]. Furthermore, the antibodies against the linear epitopes of the L1 protein during an HPV infection are also generated by and associated with previous HPV exposure, and they appear to be related to the evolution of the disease [13,14,15,16].

At first, the L1 protein was generated as a denatured protein in bacteria and used in an ELISA (Enzyme-Linked Immunosorbent Assay) to measure the anti-L1 antibodies. More recently, the Luminex immune assay has been used to evaluate several anti-L1 HPV types. Based on the results from these systems, several groups have reported a high prevalence of these antibodies in patients with low-grade lesions (~60%), which increases with the severity of the uterine cervical lesion [16,17,18]. A low prevalence of anti-L1 antibodies has also been reported among healthy women (<25%), which may point to previous HPV infections [17,18,19,20]. Recently, the generation of VLPs in baculovirus and yeast has allowed the measurement of neutralizing antibodies, as well as of antibodies against conformational epitopes [20]. These VLPs have been used in epidemiological studies to characterize the anti-VLPs immune response in different female populations, such as women from the general population, women with precancerous lesions, or those with CC [16] and, more recently, to evaluate the efficacy of the HPV vaccines [9,10,11,21]. Using the ELISA system for VLPs or denatured L1 antigens, several groups showed an antibody prevalence ranging from 7% to 43% in healthy women [14,18,21,22], from 25% to 65% in women with CIN 1–3 lesions [14,18,21,23], and from 43% to 100% in CC patients [23,24,25]. Although these results are heterogeneous, a constant was that both types of antibodies (anti-L1 and anti-VLPs) increased with the severity of the cervical lesion. However, no previous reports have evaluated the antibody response against conformational (VLPs) and linear (L1) epitopes from HPV at the same time in the same women population with different HPV exposure. Thus, we aim to understand why women with persistent HPV infection progressed to CC, even though they have generated antibodies against the L1 protein, and if any of these antibodies (anti-VLPs or anti-L1) could be useful as markers of some stage of the disease.

In this study, we used a direct ELISA to determine the serum antibody levels against linear (L1) and conformational (VLPs) epitopes from HPV16 and HPV18 (HPV16/18) in women with uterine cervical lesions or HPV-vaccinated. In turn, we also evaluated the usefulness of these anti-VLPs and anti-L1 HPV16/18 antibodies as markers to distinguish different uterine cervical lesions, as well as to differentiate from HPV-vaccinated populations (10- to 100-fold higher titers than in natural infection). Our results showed that the anti-VLPs antibodies 16/18 are type-specific and highly effective to detect vaccinated women, whereas anti-L1-16 antibodies were better to differentiate women with CIN3/CC from the study population. Our results are promising, but to validate the performance of these anti-L1-16 antibodies to differentiate high-grade lesions and CC from the general population, a larger female study group with different HPV exposure is needed.

## 2. Materials and Methods

### 2.1. Study Population

This is a retrospective study that used samples from three different serum banks corresponding to naturally infected women, women with uterine cervical lesions, and HPV-vaccinated women to characterize the antibody response against HPV16 and HPV18 VLPs (conformational epitopes) and denatured L1 (linear epitopes). The women included in the study who may have been exposed to HPV were divided into 3 groups: (1) normal adult women (*n* = 68), who had a negative Papanicolaou (Pap) test; (2) women with different degrees of cervical lesions and cervical cancer (*n* = 62), diagnosed by histopathology (15 CIN1, 8 CIN2, 10 CIN3, and 29 CC); (3) HPV-vaccinated women (*n* = 36). In the HPV-vaccinated group, 44.4% were immunized with the bivalent vaccine (*n* = 16), and 55.6% received the quadrivalent vaccine (*n* = 20). This group also self-reported a normal Pap test in the last year and no history of previous uterine cervical lesions.

Fifty serum samples from adolescent females (between 9 to 13 years of age, nonsexually active, and presumably HPV-naïve) from a serum bank were used to determine the cut-off points of the HPV16 and HPV18 ELISAs (VLPs and denatured L1 antigens).

The HPV DNA typing was performed by polymerase chain reaction (PCR) amplification on biopsies from patients, as described elsewhere [26]. Biological samples were only accessible from the normal adult women group and women with uterine cervical lesions and CC.

This research was conducted following the Declaration of Helsinki, and the protocol was evaluated and approved by the Ethical Committee at the National Institute of Public Health, Mexico (Projects No. 781 and CI 1518).

### 2.2. Production and Purification of VLPs from HPV16/18

Baculovirus HPV18-L1 (BaCU-L1-18) to produce VLPs was generated using the Bac-to-Bac Baculovirus Expression System (Invitrogen, Boston, MA, USA) by PCR amplification of the HPV18-L1 gene (1728 bp). Baculovirus HPV16-L1 (BaCU-L1-16) was provided by Dr. Martin Müller (Infections and Cancer Program, German Cancer Research Center, Heidelberg, Germany). To produce VLPs, 1 × 10^7^ High Five cells/mL were incubated for 1 h at 27 °C with 9 MOI (Multiplicity of Infection) of BaCU-L1-16 or 5 MOI of BaCU-L1-18. The cells were diluted with fresh Express Five^TM^ -SFM medium (Gibco, Gaithersburg, MD, USA) at 1 × 10^6^ cells/ mL and left to incubate with shaking for 72 h. Finally, the cells were harvested by centrifugation at 154× *g* for 5 min and washed with cold PBS (phosphate-buffered saline) to proceed with lysis and purification through a cesium chloride (CsCl) gradient, as described elsewhere [27]. To ensure the correct structure of the VLPs and to avoid the presence of denatured L1 protein in the preparations, the purified VLPs were disassembly and reassembly by treatment with 5% β-mercaptoethanol-0.15-M NaCl for 16 h at 4 °C and then dialyzed into PBS-0.5-M NaCl for 24 h at 4 °C [28]. Finally, the VLPs were aliquoted and keep at −70 °C until use.

### 2.3. Coomassie Staining and Western Blot Analysis of L1 from HPV16/18 Expressed in Baculovirus

The CsCl gradient fractions were examined for the presence of L1 protein by 10% sodium dodecyl sulfate-polyacrylamate gel electrophoresis (SDS-PAGE) and then either Coomassie-stained to verify the presence and purity of the L1 protein from HPV16/18 or treated for Western blot to corroborate the identity of L1 proteins in the preparations [29]. Proteins were transferred to PROTRAN membranes (Perkin Elmer, Waltham, MA, USA) and tested with K1H8 mouse monoclonal anti-HPV1 antibody (dil. 1:2000) (DAKO, Santa Clara, CA, USA), which recognizes a linear L1 epitope from several HPVs, including those from HPV16 and 18. Specific antibody binding was detected with goat anti-mouse antibody conjugated to horseradish peroxidase (HRP) (dil. 1:2000) (DAKO, Santa Clara, CA, USA), developed using a Western Lightning Chemiluminescence Reagent Plus kit (Perkin Elmer, Waltham, MA, USA) and exposed to X-OMAT film (Kodak, Rochester, NY, USA). The purity of the L1 protein in each fraction was calculated from the Coomassie gel by densitometric analysis using the ImageJ program (version 1.33u, NIH-USA) and took the total bands in each acrylamide gel lane as 100%. The calculated purity of the HPV16-L1 protein preparations fluctuated from 86% to 75% from top to bottom of the gradient fractions, respectively (Appendix A, lines 9–11) and the identity of the L1 protein corroborated by immunoblot (Appendix A, lines 9–11). The HPV18-L1 protein preparations presented a similar percentage of purity (Appendix A, lines 8–11). 

### 2.4. Identification of VLPs from HPV16/18 by Transmission Electron Microscopy

Aliquots of gradient fractions containing the VLPs from HPV16 or 18 were adsorbed onto copper grids covered with formvar-carbon for 1 min, contrasted with uranyl acetate for 1 min, and the excess stain removed. The grids let to air-dry for 5 min and samples analyzed in a Carl Zeiss Libra 120 Transmission Electron Microscope (TEM) (Microscopic Facility at the Instituto de Biotecnologia, UNAM, Morelos, Mexico) in search for VLP structures and microphotographs taken to evaluate the size and shape. The TEM preparations from CsCl fractions contained adequately structured and sized (~53 nm) VLPs from HPV16 or HPV18 (Appendix A, respectively), as compared to VLPs from commercial HPV vaccine (Appendix A). Denatured L1 proteins from HPV16 and 18 used for ELISA-L1 were prepared for TEM by using denaturing conditions, as described in the following section. Under these conditions, no VLPs were observed in the preparation, suggesting that the linearized L1 protein was mainly present in these preparations (Appendix A).

### 2.5. Detection of Anti-VLPs and Anti-L1 Antibodies from HPV16/18 by ELISA and Validation of the Assays

The ELISA assay was carried out with complete VLPs (diluted in PBS) to look for antibodies to conformational epitopes. In parallel, an ELISA-L1 was carried out to look for antibodies to linear L1 epitopes (VLPs irreversibly denatured with 0.2-M carbonate buffer) [28]. ELISA plates were coated with 280 ng/well of VLPs-16 or -18 in PBS or L1-16 or -18 in carbonate buffer and left for 16 h at 4 °C. Subsequently, wells were blocked with 5% lowfat milk PBS, 0.5% Tween 20, and 0.02% sodium azide for 1 h at 37 °C and the sera from women tested in duplicate at a 1:90 dilution. The antigen–antibody reaction was developed with goat anti-human antibody HRP-conjugated (dil. 1:8000) for 1 h at 37 °C and with tetramethylbenzidine (TMB) peroxidase substrate (Fitzgerald, Acton, MA, USA) (0.01% TMB and 0.003% H_2_O_2_ in 0.1-M sodium acetate, pH 6.0) for 30 min at room temperature. The reaction was stopped with 1 M H_2_SO_4_ and read at 450 nm.

The ELISA-VLPs 16/18 assay specificity was tested with a set of sera-positive for antibodies against VLPs 16 or 18 from women HPV DNA-positive for only one of the HPV types and negative against L1 protein by Western blot (denatured conditions). No antibody crossreaction between these two ELISA-VLPs 16/18 assays was observed when sera positive for anti-VLPs-16 was tested with VLPs-18 and vice versa.

For the ELISA-L1 16/18 assay specificity, we used the K1H8 monoclonal antibody (mAb) (DAKO, Santal Clara, CA, USA), which recognizes a linear L1 epitope from several HPV types, including 16 and 18, by Western blot and in the ELISA-L1 but does not recognize the conformational epitopes in the ELISA-VLPs. We used this mAb as a positive control to detect the L1 protein in ELISA-L1 and as a negative control for ELISA-VLPs, and this also allows us to evaluate the presence of the L1 protein in the VLP preparations. The anti-L1 antibody crossreaction (10–20%) to linear epitopes between different HPV types have been reported [30], and this percentage is similar to what we observed in our results.

The sensitivity and specificity of the ELISA-VLPs and ELISA-L1 HPV16/18 systems, set up in our laboratory, were validated by the receiver operating characteristic curve (ROC) analysis by using sera from HPV-vaccinated women (Appendix A), and for ELISA-VLPs HPV16/18, these parameters were similar to what has been reported previously [31,32]. The sensitivity and specificity of the ELISA-L1 were as high as for the ELISA-VLPs, although there are no previous reports for these values. Thus, we are confident that the antigens (VLPs and L1 protein) and the ELISA system for HPV16/18 cover the standard requirements to reproduce the previous results (HPV-vaccinated women) and to generate new data on the serological response of women with HPV-associated uterine cervical lesions.

The antibody titers in the ELISA system were calculated by using a four-parameter logistic equation from IVD Tools AssayFit V1.6 (Excell compatible) and expressed as ELISA units (EU)/mL. The reference curves for the anti-VLPs and anti-L1 for HPV16/18 antibodies were prepared as described by Harper et al. [10]. Briefly, previously selected serum samples positive for anti-VLPs 16/18 but negative for denatured L1 16/18 and vice versa were used for the reference titration curves. The cut-off points were calculated from the reference titration curves and used to determine the positivity of each sample as follows: 9 EU/mL for anti-VLPs-16, 41 EU/mL for anti-VLPs-18, 37 EU/mL for anti-L1-16, and 25 EU/mL for anti-L1-18.

### 2.6. Statistical Analysis

The statistical analysis was performed with the STATA 13.0 software (Stata Corp, College Station, TX, USA). Dispersion of the anti-VLPs and anti-L1 antibody levels was carried out by boxplot analyses of the different groups of females studied, and the difference between the medians (M) was calculated with Kruskal–Wallis and adjusted with the Dunnet test. Sensitivity, specificity, and areas under the ROC curve (AUC) for the two types of antibodies were calculated in the group of vaccinated women to validate the performance of the ELISA-VLPs and ELISA-L1 assays used in this study. Logistic regression analysis was used to estimate the odds ratio (OR) and 95% CIs for the association of the anti-L1 and anti-VLPs antibodies with precancerous lesions, CC, and vaccinated women, as well as with different risk factors for CC.

## 3. Results

### 3.1. Population Characteristics

To distinguish HPV-vaccinated women from naturally infected ones or from those that had developed uterine cervical lesions or CC, we evaluated the presence of HPV16 and HPV18 anti-VLPs and anti-L1 antibodies in the serum of these women. A total of 166 serum samples were analyzed: 44.7% (*n* = 68) were from adult women that reported normal Pap tests (NL, no lesion); 40.8% (*n* = 62) were from women with different cervical lesions (15 CIN1, 8 CIN2, 10 CIN3, and 29 CC); and 21.6% (*n* = 36) were from women who received the complete three-dose schedule of an HPV vaccine (Table 1).

The mean age of the study population was 37 years (range: 18–64 years), although 77.8% of the vaccinated women were in the range of 18–27 years of age, corresponding to the main age group that has received HPV vaccination in Mexico. Most of the vaccinated women were single (77.8%) and had an educational level of high school or more (100%); 41.7% were smokers. In the case of the NL women group and women that presented cervical lesions, most of them reported being married or having a steady partner (92.6% and 84%, respectively) and have a basic educational level (>75%); the majority were nonsmokers (>85%). Regarding the sexual behavior characteristics, 72.2% of the vaccinated women reported having less than seven years of sexual activity, and 50% reported having had at least two sexual partners in the last year. In contrast, the majority of the NL group and the women with lesions reported having more than 18 years of sexual activity (63.3% and 74.2%, respectively), but the number of sexual partners reported in the last year was different for these two groups (Table 1). The HPV status of the women was obtained only from the NL and the lesions women groups. The results showed that 38.2% of the NL group and 75.8% of the lesions group were positive for HPV-DNA. Detection of HPV16 was observed in 23.1% and 82.9% of the adult and lesion groups, respectively, while high-risk HPV (HR-HPV) was present in 65.4% of the adult women group and 17.1% of the lesions group (Table 1). We were unable to determine the HPV status of the HPV-vaccinated women, due to the lack of biological samples, and for this reason, we did not use this variable to carry out the statistical analysis.

### 3.2. Differential Antibody Response Against Conformational and Linear L1 Epitopes from HPV 16 and 18 in Women with CIN3/CC and Vaccinated Women

Antibody levels against the conformational and linear epitopes of the L1 protein were analyzed by ELISA-VLPs and ELISA-L1 for HPV16 and HPV18, respectively, in the serum of women with different exposures to HPV or of HPV-vaccinated women. The analysis showed that the HPV-vaccinated women presented higher anti-VLPs-16 and anti-VLPs-18 antibody levels (M = 68,721.6 EU/mL and M = 2028.3 EU/mL, respectively) than the NL group (*p* < 0.0001) (Figure 1A,B, left side of the graphs). It was also observed that the antibody levels in the vaccinated group were one log higher for anti-VLPs-16 than for anti-VLPs-18 (Figure 1A,B, left side of the graphs). When we analyzed the lesion groups, a statistical difference in the anti-VLPs-18 antibody levels was observed only with the CIN3/CC group (*p* < 0.05) (Figure 1B, left side of the graph).

Similar results were observed when anti-L1 antibody levels were measured, since the vaccinated women showed higher anti-L1-16 (M = 5081.4 EU/mL) and anti-L1-18 antibody levels (M = 193.9 EU/mL) than the NL group (*p* < 0.05) (Figure 1A, B, the right side of the graphs). Statistical differences regarding the anti-L1-16 antibody levels were only observed between the CIN1-2 and the NL groups (Figure 1A, the right side of the graph).

### 3.3. Differential Association of Antibodies against L1 from HPV16/18 with the Presence of CIN3/CC and HPV Vaccination 

To investigate whether the exposure to the HPV infection or the vaccination was associated with anti-VLPs (conformational epitopes) and anti-L1 (linear epitopes) antibodies, a logistic regression analysis with these antibodies and other behavioral characteristics was carried out (Table 2). The analysis showed that anti-VLPs-16/18 antibodies were highly associated with the HPV-vaccinated women (OR = 2.11 × 10^8^ and 16.50, respectively) (*p ≤* 0.001) and, to a lesser extent, with the CIN3/CC group (OR = 2.29, for anti-VLPs-16), albeit without statistical significance (Table 2). On the other hand, the analysis of the behavioral characteristics showed that the women that reported having two or more sexual partners in the last year were associated with the anti-VLPs-16 antibodies (OR = 2.09, *p* = 0.032), whereas smoking was associated with anti-VLPs-16 (OR = 3.99) and anti-VLPs-18 antibodies (OR = 3.88), and this was statistically significant (*p <* 0.002 and 0.005, respectively) (Table 2).

On the other hand, the regression analysis of anti-L1 antibodies in the different groups showed high associations of both anti-L1-16 and anti-L1-18 antibodies with the HPV-vaccinated women (OR = 536.0 and 49.2, respectively); these associations were statistically significant (*p* = 0.000). Besides, an association between these antibodies and the group of women with CIN3/CC was only observed with anti-L1-16 (OR = 12.18), and this was statistically significant (*p* = 0.023) (Table 2). The association analysis between the anti-L1 16/18 antibodies and the behavioral characteristics showed that both anti-L1 antibodies were associated with women that reported two or more partners in the last year (OR = 2.20 and 1.62, respectively), only being significant for anti-L1-16 (*p* = 0.032). In the case of women that smoked, high associations were observed for anti-L1-16 and -18 antibodies (OR = 5.30 and 4.84, respectively), and those were highly significant (*p* = 0.000 and 0.001, respectively) (Table 2).

Since a positive association was observed between anti-L1-16 antibodies and the CIN3/CC group, we carried out a ROC analysis by using the regression model for the HPV-exposed women groups only (not considering HPV-vaccinated women) to evaluate the performances of the anti-L1-16 antibodies to distinguish the cases from the general population. The results showed that the anti-L1-16 antibodies presented a moderate performance to distinguish the CIN3/CC group (AUC = 0.63) with high specificity (98.5%) and low sensitivity (15.4%), but the assay correctly classified 93.5% of the subjects (data not shown). The low sensitivity of the anti-L1-16 assay could be due to the small number of subjects in the group and because the analysis was not restricted to HPV DNA status.

It was not possible to carry out the association analysis for the different 16/18 antibodies with the variables of age and time of sexual activity, because the youngest age group (18–27 years old) was used as the reference group for the analysis, and 72.8% of the HPV-vaccinated women corresponded to this group. 

## 4. Discussion

In this study, we aim to understand why women with persistent HPV infection progressed to CC, even though they generated antibodies against the L1 protein, and if any of these antibodies could be useful as markers of stage disease. To achieve this, we looked for antibodies against conformational epitopes (VLPs) as an indirect measure of neutralizing activity, as has been previously demonstrated by Dessy et al. [9], and for antibodies against linear epitopes (L1) as a measure of HPV exposure [13,16]. Notably, no previous reports measured both anti-VLPs and anti-L1 antibodies in the same population of women and determined the differences in the antibody responses between women who are naturally infected and with uterine cervical lesions from those that are HPV-vaccinated. 

From different studies, contradictory results have been reported when anti-VLPs and anti-L1 antibodies were evaluated in different populations. For instance, in the Wang SS and colleagues study [33], it was shown that the anti-VLPs HPV16/18 antibodies were strongly associated with the HPV DNA-positive CIN3/cancer women group (OR = 34.7 and 16.6, respectively), but the associations dropped drastically when only the presence of anti-VLPs HPV16/18 antibodies were considered in the statistical analysis (OR = 2.0 and 1.9, respectively) [33]. In contrast, Urquiza and colleague’s study [18] showed that antibodies against three L1 peptides from HPV16 were more sensitive than anti-VLPs antibodies for the detection of HPV DNA-positive cervical lesions [18]. Our results are in agreement with both studies, as we observed a similar low association for anti-VLPs-16 antibodies in the CIN3/CC group (OR = 2.29), as in the Wang report when the HPV DNA status was not considered in the analysis. Although this association was not significant in our study, this result argues in favor that the ELISA-VLPs assay, set up in our laboratory, is confident and able to reproduce other group results. At the same time, our results also agree with the Urquiza report, as we also found that antibodies against the L1 protein showed a higher association with anti-L1-16 antibodies (OR = 12.18) than with anti-VLPs-16 antibodies (OR = 1.82) for the CIN3/CC group. These results suggest that anti-L1-16 antibodies are better markers to differentiate women with high-grade lesions or CC with high specificity (98.5%), although the sensibility of the system needs to be increased by testing a larger women population. 

Moreover, our combined ELISA-VLP and –L1 assays also showed a differential antibody response against conformational and linear epitopes (VLPs and L1, respectively), as the antibody levels against HPV16/18 VLPs were higher (two and one log, respectively) among vaccinated women than among naturally infected women, and this is in agreement with what has been reported previously for anti-VLPs antibodies [34]. However, no information about the presence of anti-L1 antibodies in vaccinated women has been reported before, which suggests that linear epitopes of the L1 antigen could be present on the VLPs, and those generate the anti-L1 antibodies, but this has to be further analyzed.

The performance of anti-VLPs antibodies to detect uterine cervical lesions has been analyzed previously. Coseo and colleagues [35] reported that the ELISA-VLP assay showed moderate power (AUC = 0.7 for HPV16/18) to detect HPV-DNA-positive cases and that this slightly increased its performance for HPV16 when abnormal cytology was considered (≥CIN2+). In contrast, Gonçalves et al. [36] reported that the ELISA-VLP assay showed a good correlation between the detection of anti-VLPs antibodies and the presence of HPV16/18 infection. Our results differ from these reports, since we observed that HPV16/18 anti-VLPs antibodies were not able to distinguish uterine cervical lesions. However, the anti-L1-16 antibodies displayed a moderate performance to differentiate positive CIN3/CC cases (AUC = 0.63) from the population studied with high accuracy (PV = 93.5%). The performance of the assay could be improved by increasing the number of subjects in the lesion group and by including the HPV status in the analysis.

Another explanation for the discrepancy with the Coseo results [35] could be the different protocols used in the preparation of the VLP antigens for the ELISA assays [10,35]. The antigen diluent usually used for the ELISA-VLPs is PBS with a neutral pH 7 and for L1 preparations a high pH–carbonate buffer assay is required, as has been reported before [28,37], and as we showed with the characterization by TEM of the VLPs and L1 generated in our laboratory. In our study, PBS and carbonate buffer treatments of the VLPs were used, as we wanted to differentiate the antibody response against conformational epitopes (VLPs) from that against linear epitopes (denatured L1) in women with different uterine cervical lesions.

On the other hand, it has been reported that lower levels of anti-VLPs-18 antibodies compared to anti-VLPs-16 antibodies are commonly found among vaccinated women [38]. This low antibody response has probably been associated with the reduced immunogenicity of HPV18 VLPs [39]. Our results are in agreement with that study, as we identified one log higher levels of anti-VLPs-16 antibodies (M = 68,721 EU/mL) compared to against anti-VLPs-18 antibodies (M = 2028 EU/mL) in the vaccinated group, pointing to the hypothesis of the low immunogenicity of HPV18 VLPs [39].

There are still limitations for the ELISA test, especially regarding its sensitivity to detect VLPs or L1 antigens. This is partly because the seroconversion period during a natural HPV infection differs between women (from six to more than 12 months), and the antibody titers do as well. Several research groups have identified a heterogeneous anti-VLPs and anti-L1 antibody prevalence in different female populations. For instance, by using and ELISA-VLPs for HPV16, the prevalence range was from 7% to 43% in healthy women, from 25% to 68% among women with CIN lesions, and from 28% to 100% among women with CC [14,15,21,23,40]. When the L1 protein from HPV16 was used in the ELISA (linear L1 or recombinant GST-L1), the prevalence varied from 3% to 52% in healthy women, from 10% to 95% in CIN lesions, and from 21% to 100% in CC [17,18,41,42]. The variability in the ELISA results could be due to differences in the antigen preparations (VLPs or L1) used by each laboratory. In the case of using the recombinant L1 antigen, contamination with bacterial proteins may hinder the sensitivity of the system due to high backgrounds. In the case of the VLP antigen, stability and conservation of the VLP structure due to preservation buffer and freezing conditions may generate low specificity of the system. In our system, to avoid contamination with the denatured L1 protein in the VLP preparations, we disassembled and reassembled the VLPs by the β-mercaptoethanol treatment and ensured the separate measurements of the anti-VLPs and anti-L1 antibodies.

Therefore, a standardized procedure to carried out the ELISA assays for VLPs and L1 needs to be established to correctly evaluate the protective immune responses of the HPV vaccine (anti-VLPs antibodies as surrogate markers for neutralizing antibodies) and, also, to clarify the association of the anti-L1-16 antibodies as surrogate markers of high-grade uterine cervical lesions.

## 5. Conclusions

Overall, our results suggest that the anti-VLPs 16/18 antibodies are highly effective to detect vaccinated women (100% and 61.1%, respectively), but the anti-L1 16 antibodies are better markers to distinguish women with CIN3/CC in the general population. However, to validate the performance of this ELISA-L1 for HPV16, the HPV status of the women should be included, and a larger population study should be carried out.

## Figures and Tables

**Figure 1 vaccines-09-00442-f001:**
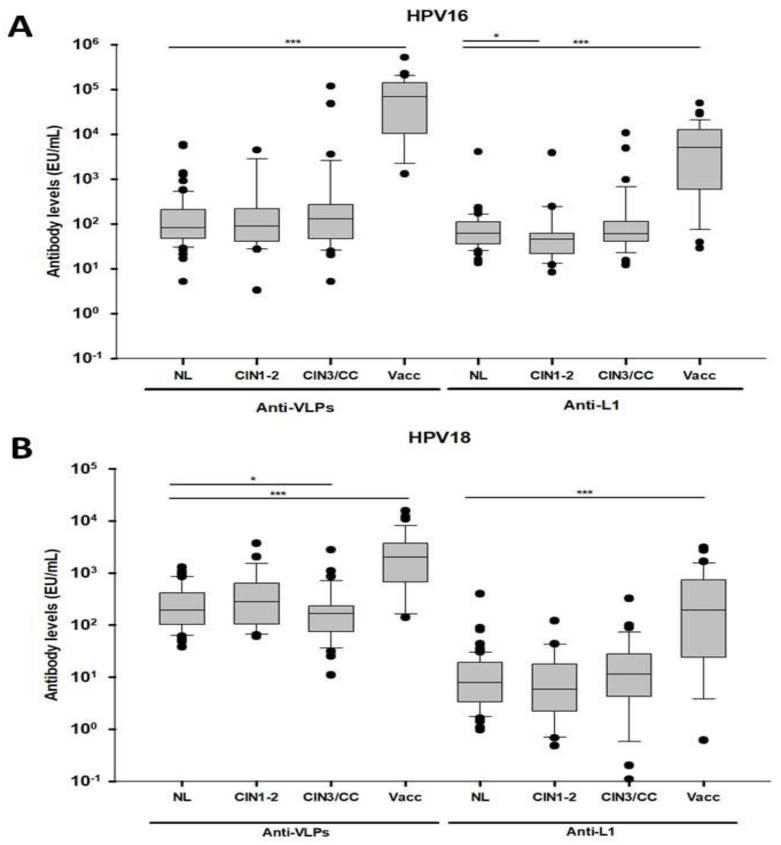
Antibody response against VLPs and L1 from HPV16 and HPV18. The distributions of the antibody levels (EU/mL) obtained from the VLPs and L1 from HPV16 (**A**) and HPV18 (**B**) by ELISA were plotted for each group studied and represented as data scatter boxes. The EUs for all the samples were adjusted for the dilution factor (dil 1:100). NL, no lesion; CIN1-2, cervical intraepithelial neoplasia 1–2; CIN3/CC, cervical cancer; Vacc, vaccinated women. Statistical significance * *p* < 0.05 and *** *p* < 0.0001.

**Table 1 vaccines-09-00442-t001:** Demographic and sexual behavior characteristics of the study population.

Characteristic		No Lesion		Lesions		Vaccinated	
Demographic		*n* = 68	%	*n* = 62	%	*n* = 36	%
**Age**							
18–27 years		8	11.8	4	6.5	28	77.8
28–37 years		19	27.9	13	21.0	5	13.9
≥38 years		41	60.3	45	72.5	3	8.3
	*Total*	68		62		36	
**Marital status**							
*Single*		1	1.5	5	8.0	28	77.8
*Married or steady partner*		63	92.6	52	84.0	8	22.2
*Widow, Divorced or separated*		4	5.9	5	8.0	0	0.0
*Total*		68		62		36	
**Educational level**							
*Basic*		52	76.5	28	77.8	0	0.0
≥*High School*		11	16.2	2	5.5	36	100.0
*No education*		5	7.3	6	16.7	0	0.0
*Missing*		-	-	26	-	-	-
	*Total*	68		62		36	
**Smoking**							
*No*		59	86.8	32	88.9	21	58.3
*Yes*		9	13.2	4	11.1	15	41.7
*Missing*		-	-	26	-	-	-
	*Total*	68		62		36	
**Sexual Behavior**							
**Time of sexual activity ^1^**							
≤7 years		6	8.8	4	6.4	26	72.2
8–17 years		19	27.9	12	19.4	7	19.4
18–27 years		25	36.8	30	48.4	1	2.8
≥28 years		18	26.5	16	25.8	2	5.6
	*Total*	68		62		36	
**Number of sexual partners last year**							
0–1		63	92.7	18	29.0	18	50.0
≥2		5	7.3	44	71.0	18	50.0
	*Total*	68		62		36	
**Diagnostic (Pap or histopathology)**							
*Normal*		68	100	-	-	-	-
*CIN1*		-	-	15	24.2	-	-
*CIN2*		-	-	8	12.9	-	-
*CIN3*		-	-	10	16.1	-	-
*CC*		-	-	29	46.8	-	-
	*Total*	68		62			
**HPV STATUS**							
*Negative*		42	61.8	15	24.2	-	-
*Positive*		26	38.2	47	75.8	-	-
**HPV genotypes**							
*HPV16*		6	23.1	39	82.9	-	-
*HR-HPV*		17	65.4	8	17.1	-	-
*LR-HPV*		3	11.5	0	0.0	-	-

^1^ Obtained by subtracting the age of onset of sexual life to the age at the time of the study.

**Table 2 vaccines-09-00442-t002:** Differential association of antibodies against VLPs and L1 from HPV16 and 18 in women with uterine cervical lesions and HPV-vaccinated.

**Anti-VLPs Seropositivity**											
**Variable**	**Total**	**HPV16 (*n* = 49)**					**HPV18 (*n* = 25)**				
	***n***	***n***	**(%)**	**OR**	**(CI 95%)**	***p***	***n***	**(%)**	**OR**	**(CI 95%)**	***p***
Population Group											
Normal Adult	68	5	(7.4)	1.00			0	(−)			
CIN1-2	23	2	(8.7)	1.20	(0.21–6.65)	0.835	2	(8.7)	1.00 ^3^		
CIN3/CC	39	6	(15.4)	2.29	(0.65–8.07)	0.197	1	(2.6)	0.27	(0.23–3.23)	0.305
Vaccinated	36	36	(100.0)	2.11 × 10^8^		0.000	22	(61.1)	16.50	(3.33–81.53)	0.001
No. sexual partners last year											
0–1	99	23	(23.2)	1.00			14	(14.1)	1.00		
≥2	67	26	(38.8)	2.09	(1.06–4.12)	0.032	11	(16.4)	1.19	(0.50–2.81)	0.688
Smoking ^1^											
No	112	28	(25.0)	1.00			14	(12.5)	1.00		
Yes	28	16	(57.1)	3.99	(1.68–9.47)	0.002	10	(35.7)	3.88	(1.49–10.10)	0.005
**Anti-L1 Seropositivity**											
**Variable**	**Total**	**HPV16 (*n* = 40)**					**HPV18 (*n* = 32)**				
	***n***	***n***	**(%)**	**OR**	**(CI 95%)**	***p***	***n***	**(%)**	**OR**	**(CI 95%)**	***p***
Population Group ^2^											
Normal Adult	68	1	(1.5)	1.00			3	(4.4)	1.00		
CIN1-2	23	1	(4.4)	3.04	(0.18–50.75)	0.438	1	(4.4)	0.98	(0.97–9.96)	0.990
CIN3/CC	39	6	(15.4)	12.18	(1.40–105.38)	0.023	3	(7.7)	1.80	(0.34–9.41)	0.483
Vaccinated	36	32	(88.9)	536.0	(57.55–4991.7)	0.000	25	(69.4)	49.2	(12.67–191.3)	0.000
No. sexual partners last year											
0–1	99	18	(18.2)	1.00			16	(16.1)	1.00		
≥2	67	22	(32.8)	2.20	(1.06–4.52)	0.032	16	(23.9)	1.62	(0.74–3.53)	0.219
Smoking ^1^											
No	112	20	(17.9)	1.00			17	(15.2)	1.00		
Yes	28	15	(53.6)	5.30	(2.18–12.87)	0.000	13	(46.4)	4.84	(1.96–11.96)	0.001

^1^ There are 26 missing data. ^2^ AUC = 0.63 for the logistic regression model for anti-L1-16 antibodies of the studied population without considering the vaccinated group. ^3^ The CIN1-2 group was used as a reference for the regression model.

## Data Availability

All the relevant data are accessible within the article and in the Appendix A.

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
