# Peer review of "Differential Antibody Response against Conformational and Linear Epitopes of the L1 Proteins from Human Papillomavirus Types 16/18 Is Observed in Vaccinated Women or with Uterine Cervical Lesions"

_vaccines, 2021, doi:10.3390/vaccines9050442_

Round 1

Reviewer 1 Report

In this manuscript, Pedroza-Saavedra et al. describe a study, in which they tried to determine the antibody response against VLPs and linear epitopes of HPV 16/18 L1 proteins in different populations, including HPV-vaccinated women and those with uterine cervical lesions. They found the anti-VLPs 16/18 antibodies are highly effective 395 to detect vaccinated women, whereas the anti-L1 16 antibodies are better to distinguish the women with CIN3/CC from other populations. This study, which detected both anti-VLP and anti-L1 antibody responses in the same population, is interesting and is very useful for evaluating the effect of vaccination and cancer therapy. However, the lacking the detailed information of HPV infection in clinical cases makes the conclusion is still questionable.  Some specific comments are listed below.

  1. In the study population, some situations may affect the detection of antibody response. For example, do HPV-vaccinated women have no HPV 16/18 infection? Or, are HPV-vaccinated women with no uterine cervical lesions?
  2. The information of the vaccines applied in HPV-vaccinated populations should be given.   
  3. In this detect system, could HPV 16 VLP or L1 protein crossreact with HPV 18 antibodies or vice versa?
  4. Does the case with a higher titer of HPV 16/18 VLP or L1 antibodies have HPV 16 /18 infection?

Author Response

Response to Reviewer 1 comments

Reviewer comment:

In this manuscript, Pedroza-Saavedra et al. describe a study, in which they tried to determine the antibody response against VLPs and linear epitopes of HPV 16/18 L1 proteins in different populations, including HPV-vaccinated women and those with uterine cervical lesions. They found the anti-VLPs 16/18 antibodies are highly effective 395 to detect vaccinated women, whereas the anti-L1 16 antibodies are better to distinguish the women with CIN3/CC from other populations. This study, which detected both anti-VLP and anti-L1 antibody responses in the same population, is interesting and is very useful for evaluating the effect of vaccination and cancer therapy. However, the lacking the detailed information of HPV infection in clinical cases makes the conclusion is still questionable. Some specific comments are listed below.

Point 1: In the study population, some situations may affect the detection of antibody response. For example, do HPV-vaccinated women have no HPV 16/18 infection? Or, are HPV-vaccinated women with no uterine cervical lesions?

Response 1:

Several reports have shown that antibody induction by the HPV vaccine is higher (10 to 100 fold) than the one produced by natural exposure to the virus (Stanley, et al 2012; Schiller & Müller, 2015). Also, it is well documented that Mexican population the prevalence of HPV16 and HPV18 is 4.13% and 1.7%, respectively (Campos-Romero et al., 2019). Thus, it is unlikely that the high titers of anti-HPV antibodies generated by the HPV vaccine and the low prevalence of HPV16/18 in the Mexican population (Vaccinated group n=36, < 2 women) could affect the results observed in our study.    

On the other hand, the HPV vaccinated women that participated in the original study self-reported a normal Pap test in the last year and no history of previous uterine cervical lesions. For this reason, we did not consider this factor to affect the antibody response. To clarify this important data that the reviewer points out, this information was added in the Material and Methods section, 2.1 Study population (1st paragraph, lines 10-11) and showed in a bold letter:

 HPV vaccinated women self-reported a normal Pap test in the last year and no history of previous uterine cervical lesions.”

Point 2: The information of the vaccines applied in HPV-vaccinated populations should be given.   

Response 2:  

The information of the vaccines administrated to the women that were selected from the serum bank was not available to the research group until the end of the study. We have the information now and this included 16 women immunized with the bivalent vaccine and 20 women with the quadrivalent vaccine. The information of the specific number of women vaccinated with each type of vaccine has been added in the Material and Methods section, 2.1 Study population (1st paragraph, lines 9-10) and follows and showed in a bold letter:

“In the HPV-vaccinated group, 44.4% were immunized with the bivalent vaccine (n= 16) and 55.6% received the quadrivalent vaccine (n= 20).”

Point 3: In this detect system, could HPV 16 VLP or L1 protein crossreact with HPV 18 antibodies or vice versa?

Response 3:

To identify the specificity of the system and to determine the crossreaction in the ELISA-VLPs HPV16 versus HPV18 system, we previously searched in the serum bank for samples of sera from women who tested positive for HPV16 or 18 DNA, and analyze those sera for the presence of specific anti-VLPs antibodies in the ELISA assay, and negative by Western blot under denatured conditions. In this way, we were certain that the sera only recognized the VLPs. Then, we tested the selected positive anti-VLPs-16 sera in the ELISA-VLPs HPV18 and the positive anti-VLP-18 sera in the ELISA-VLPs HPV16 and found no crossreaction of the antibodies in either system.

For the ELISA-L1 system, we used the K1H8 monoclonal antibody (mAb) from the DAKO Company (USA), which recognizes a linear L1 epitope from several HPV types including 16 and 18, which suggested that antibody crossreaction exist. In a separate study, we generated rabbit polyclonal antibodies against L1 protein from HPV16 and 18 and evaluated the crossreaction between them. By using ELISA and Western blot assays for L1 proteins, we determined that the anti-L1-16 polyclonal antibodies crossreact up to 18% with the L1-18 protein and a similar percentage (15%) was observed when we tested the anti-L1-18 polyclonal antibodies with the L1-16 protein (unpublished results). From the population studied, we identified that 82.9 % was HPV16 DNA positive and 70 % presented CIN3/CC lesions, and only one woman was HPV18 DNA positive. Thus, the ELISA-L1 assay is not type-specific, and the 21.9% (7/32) of anti-L1-18 antibodies observed in our population was probably the result of crossreaction. Although this crossreaction did not affect our results, when the logistic regression analysis was performed, only association with anti-L1-16 antibodies was observed.

Point 4: Does the case with a higher titer of HPV 16/18 VLP or L1 antibodies have HPV 16 /18 infection?

Response 4:

As we described in the previous point, the ELISA-VLPs assay is highly specific for the HPV type and no crossreaction was observed. When we analyzed carefully the antibody titers, as the reviewer suggests, we observed that the highest anti-VLPs-16 antibody titers were present in 50% of women with CC and HPV16 positives and 50% of NL women who were negative for HPV. For the anti-L1-16 antibodies, the highest titers were observed in CIN lesions and CC, and all of them were HPV16 positive. However, for anti-VLPs-18 antibodies, the higher titers were observed in women with CC, CIN1, and NL but only the CC case was positive for HPV16. A similar result was observed for anti-L1-18 antibodies where the highest titers were observed in CC and NL women, and the CC case was HPV16 positive.

These observations corroborate that the HPV16 ELISA-VLPs and L1 are highly specific systems to detect this specific HPV type and differentiate between HPV-vaccinated individuals from natural infection, as has been reported before (Petrosky, et al., 2015).  The new finding is that the ELISA-L1-16 system is good to detect anti-L1-16 antibodies and those were associated with high-grade lesions. However, the low prevalence of HPV18 infection in the population studied makes it difficult to evaluate the utility of the anti-L1-18 antibodies in the detection of high-grade lesions as was observed with the anti-L1-16 antibodies. To test this hypothesis a larger sample size will be necessary to increase the number of HPV18 positive women.

References:

Stanley, M., Pinto, L. A., & Trimble, C. (2012). Human papillomavirus vaccines--immune responses. Vaccine, 30 Suppl 5(SUPPL.5), F83-7. doi.org/10.1016/j.vaccine.2012.04.106

Schiller, J. T., & Müller, M. (2015). Next generation prophylactic human papillomavirus vaccines. The Lancet. Oncology, 16(5), e217-25. doi.org/10.1016/S1470-2045(14)71179-9

Campos-Romero, A.,  Anderson, K. S., Longatto-Filho, A., Luna-Ruiz Esparza,M. A., Morán-Portela, D. J., Castro-Menéndez, J. A., et al. The burden of 14 hr-HPV genotypes in women attending routine cervical cancer screening in 20 states of Mexico: a cross-sectional study. Sci Rep. 2019; 9: 10094.

Petrosky, E- Y., Hariri, S., Markowitz, L. E., Panicker, G., Unger, E. R., Dunne E. F. Is vaccine-type seropositivity a marker for human papillomavirus vaccination? National Health and Nutrition Examination Survey, 2003–2010. Int J Infect Dis. Int J Infect Dis. 2015; 33: 137–141.  doi: 10.1016/j.ijid.2015.01.

Reviewer 2 Report

Overall, this manuscript is well written and clear. The authors sought to assess antibodies in women who have been vaccinated against HPV and women who have cervical intraepithelial neoplasia or cervical cancer. To do this, they obtained samples from women with different CIN/CC status, healthy women with a negative Pap test, control young girls (9-13) assumed to not have previous sexual activity, and HPV vaccinated women. Then ELISA was carried out and the magnitude of the antibody response detected against whole HPV VLPs (detecting antibodies to conformational epitopes) or denatured L1 protein (detecting antibodies to linear epitopes). Various statistical comparisons were made. The authors show that HPV vaccinated individuals tended to make antibodies to both linear and conformational epitopes, while only CIN3/CC women were more likely to have antibodies to L1 linear epitopes. This had high specificity but low sensitivity.

My only concern was needing more context for why these experiments were done. The authors did a good job of explaining the previous literature, but I wasn't sure what dispute or outstanding question was really being answered with this analysis. Are the authors looking for a diagnostic for CIN3/CC? Are they looking to be able to distinguish between vaccinated and HPV infected women? The authors should clarify in the introduction and the discussion what the goal or hypothesis of the work was.

Author Response

Response to Reviewer 2 comments

Reviewer comment:

Overall, this manuscript is well written and clear. The authors sought to assess antibodies in women who have been vaccinated against HPV and women who have cervical intraepithelial neoplasia or cervical cancer. To do this, they obtained samples from women with different CIN/CC status, healthy women with a negative Pap test, control young girls (9-13) assumed to not have previous sexual activity, and HPV vaccinated women. Then ELISA was carried out and the magnitude of the antibody response detected against whole HPV VLPs (detecting antibodies to conformational epitopes) or denatured L1 protein (detecting antibodies to linear epitopes). Various statistical comparisons were made. The authors show that HPV vaccinated individuals tended to make antibodies to both linear and conformational epitopes, while only CIN3/CC women were more likely to have antibodies to L1 linear epitopes. This had high specificity but low sensitivity.

Point 1: My only concern was needing more context for why these experiments were done. The authors did a good job of explaining the previous literature, but I wasn't sure what dispute or outstanding question was really being answered with this analysis. Are the authors looking for a diagnostic for CIN3/CC? Are they looking to be able to distinguish between vaccinated and HPV infected women? The authors should clarify in the introduction and the discussion what the goal or hypothesis of the work was.

Response 1:

Recently, we reviewed the literature for studies of the humoral immune response against HPV as a source of biomarkers for the prediction and detection of CC (Gutierrez-Xicotencatl et al., 2016). In this analysis, we identified different studies that have looked for the presence of anti-VLPs and anti-L1 antibodies in women with different uterine cervical lesions and CC, but no clear conclusions about the utility of these antibodies to discriminate the infection stage or the progression of the disease have been assayed. None of the studies have evaluated the anti-VLPs and anti-L1 antibodies at the same time in the same population. Thus, we were interested in understanding why women with persistent HPV infection progressed to CC, even though they have generated antibodies against the L1 protein, and if any of these antibodies (anti-VLPs or anti-L1) could be useful as markers of some stage of the disease. 

To evaluate HPV neutralizing antibodies we used the ELISA-VLPs, which is the most suitable system to evaluate large populations, highly sensible, and technically easy to standardized. Besides, with the ELISA-VLPs is possible to detect antibodies against conformational epitopes, which was shown to highly correlate with neutralizing antibodies evaluated by another type of assays, such as the competitive Luminex Immune-assay (cLIA), the pseudovirion-based neutralization assay (PBNA), or the hemagglutination assay (HA) (Roden, et al. 1996; Opalka, et al., 2003; Dias, et al., 2005; Dessy, et al., 2008). However, we were unsure that these ELISA-VLPs only detected neutralizing epitopes, and also whether non-neutralizing antibodies (against linear epitopes) were indirectly determined during the assays. A mixture of VLPs and L1 protein could be present in the antigen preparation, and because of that, inconclusive results were observed with the infection stage and the disease progression. 

To clarify this important point, we have added the central hypothesis of our work in the introduction and the discussion section and showed in a bold letter:

Introduction

Page 2, 3rd paragraph, lines 19-24:

“However, no previous reports have evaluated the antibody response against conformational (VLPs) and linear (L1) epitopes from HPV at the same time in the same women population with different HPV exposure. Thus, we aim to understand why women with persistent HPV infection progressed to CC, even though they have generated antibodies against the L1 protein, and if any of these antibodies (anti-VLPs or anti-L1) could be useful as markers of some stage of the disease.”

Page 2, 4th paragraph, lines 3-6:

In turn, we also evaluated the usefulness of these anti-VLPs and anti-L1 HPV16/18 antibodies as markers to distinguish different uterine cervical lesions as well as to differentiate from HPV-vaccinated populations (10 to 100 fold higher titers than in natural infection).”

Discussion;

Page 10, 1st paragraph, lines 1-3:

“In this study, we aim to understand why women with persistent HPV infection progressed to CC, even though they have generated antibodies against the L1 protein, and if any of these antibodies could be useful as markers of stage disease.” To achieve this, …..”

References:

Gutierrez-Xicotencatl, L.; Salazar-Pina, D.A.; Pedroza-Saavedra, A.; Chihu-Amparan, L.; Rodriguez-Ocampo, A.N.; Maldonado-Gama, M.; Esquivel-Guadarrama, F.R. Humoral Immune Response Against Human Papillomavirus as Source of Biomarkers for the Prediction and Detection of Cervical Cancer. Viral Immunol 2016, 29, 83-94. doi:10.1089/vim.2015.0087.

Roden, R. B., Hubbert, N. L., Kirnbauer, R., Christensen, N.D., Lowy, D. R., Schiller, J.T. Assessment of the serological relatedness of genital human papillomaviruses by hemagglutination inhibition. J Virol. 1996 May; 70(5): 3298–3301.

Opalka, D.; Lachman, C.E.; MacMullen, S.A.; Jansen, K.U.; Smith, J.F.; Chirmule, N.; Esser, M.T. Simultaneous quantitation of antibodies to neutralizing epitopes on virus-like particles for human papillomavirus types 6, 11, 16, and 18 by a multiplexed luminex assay. Clin Diagn Lab Immunol 2003, 10, 108-115, doi:10.1128/cdli.10.1.108-115.2003.

Dias D, Doren J Van, Schlottmann S, et al. Optimization and Validation of a Multiplexed Luminex Assay To Quantify Antibodies to Neutralizing Epitopes on Optimization and Validation of a Multiplexed Luminex Assay To Quantify Antibodies to Neutralizing Epitopes on Human Papillomaviruses 6,11,16,18. 2005;12(8):959-969. doi:10.1128/CDLI.12.8.959

Dessy FJ, Giannini SL, Bougelet CA, Kemp TJ, David MP, Poncelet SM, Pinto LA, Wettendorff MA. Correlation between direct ELISA, single epitope-based inhibition ELISA and pseudovirion-based neutralization assay for measuring anti-HPV-16 and anti-HPV-18 antibody response after vaccination with the AS04-adjuvanted HPV-16/18 cervical cancer vaccine. Hum Vaccin. 2008; 4(6): 425-34.

Round 2

Reviewer 1 Report

 The changes made significantly improve this manuscript.  It has reached the level of publication. ​